# Synergistic Effects on Incorporation of β-Tricalcium Phosphate and Graphene Oxide Nanoparticles to Silk Fibroin/Soy Protein Isolate Scaffolds for Bone Tissue Engineering

**DOI:** 10.3390/polym12010069

**Published:** 2020-01-02

**Authors:** Fan Liu, Chen Liu, Bowen Zheng, Jia He, Jun Liu, Cen Chen, In-seop Lee, Xiaohong Wang, Yi Liu

**Affiliations:** 1Center of 3D Printing & Organ Manufacturing, School of Fundamental Sciences, China Medical University (CMU), No. 77 Puhe Road, Shenyang North New Area, Shenyang 110122, China; liufan-sky@163.com (F.L.); 18856152351@163.com (C.L.); 2Department of Orthodontics, School of Stomatology, China Medical University, Shenyang 110122, China; zhengbowen1991@126.com (B.Z.); hejia_777@163.com (J.H.); liu18240293491@163.com (J.L.); 3College of Life Sciences, Zhejiang Sci-Tech University, Hangzhou 310018, China; ci0_0ci@hotmail.com; 4Institute of Natural Sciences, Yonsei University, Seoul 120-749, Korea; inseop@yonsei.ac.kr; 5Center of Organ Manufacturing, Department of Mechanical Engineering, Tsinghua University, Beijing 100084, China

**Keywords:** graphene oxide, β-tricalcium phosphate, silk fibroin, soy protein isolate, scaffold, osteogenesis

## Abstract

In bone tissue engineering, an ideal scaffold is required to have favorable physical, chemical (or physicochemical), and biological (or biochemical) properties to promote osteogenesis. Although silk fibroin (SF) and/or soy protein isolate (SPI) scaffolds have been widely used as an alternative to autologous and heterologous bone grafts, the poor mechanical property and insufficient osteoinductive capability has become an obstacle for their in vivo applications. Herein, β-tricalcium phosphate (β-TCP) and graphene oxide (GO) nanoparticles are incorporated into SF/SPI scaffolds simultaneously or individually. Physical and chemical properties of these composite scaffolds are evaluated using field emission scanning electron microscope (FESEM), X-ray diffraction (XRD) and attenuated total reflectance Fourier transformed infrared spectroscopy (ATR-FTIR). Biocompatibility and osteogenesis of the composite scaffolds are evaluated using bone marrow mesenchymal stem cells (BMSCs). All the composite scaffolds have a complex porous structure with proper pore sizes and porosities. Physicochemical properties of the scaffolds can be significantly increased through the incorporation of β-TCP and GO nanoparticles. Alkaline phosphatase activity (ALP) and osteogenesis-related gene expression of the BMSCs are significantly enhanced in the presence of β-TCP and GO nanoparticles. Especially, β-TCP and GO nanoparticles have a synergistic effect on promoting osteogenesis. These results suggest that the β-TCP and GO enhanced SF/SPI scaffolds are promising candidates for bone tissue regeneration.

## 1. Introduction

Bone deformities from congenital deformity, traumatic injury, and oncologic resection severely affect patients’ physical function and mental health. Although autologous, allograft, or xenograft bone transplantations can repair dysfunctional or defect bones in clinic, many limitations still need to be addressed: transplanted bone infection or instability, insufficient transplanted volume, even immunological rejection [1,2,3]. Thus, bone tissue engineering, involving the fabrication of three-dimensional porous scaffolds and seeding osteogenesis cells with biologically active factors on the porous scaffolds, has merged as a solution to replace traditional bone repair methods [4]. Among them, hybrid biodegradable scaffolds are of increasing interest as a strategy in recent years.

Silk fibroin (SF), a protein derived from Bombyx mori silkworm cocoons, is a promising natural polymer that has excellent biocompatibility and controllable degradability [5,6]. Although SF scaffolds have been reported to support mesenchymal stem cell (MSC) attachment, proliferation, and extracellular matrix (ECM) deposition [7], it is insufficient to induce stem cell differentiation, and even repair large bone defects. Thus, blending of SF with other polymers is expected to develop double-network composite scaffolds that enhance biological properties [8]. Among them, soy protein isolate (SPI), a dietary protein extracted from the soy bean, has emerged as an attractive alternative to animal-derived protein source for biomedical applications. It contains various bioactive peptides, and has been approved by the Food and Drug Administration (FDA) of the United States for its potential health benefits. SPI could significantly improve trabecular number, bone volume, and bone mineral density in mice [9,10]. More importantly, some researches indicated that SPI might counteract the detrimental effects of osteoporosis and obesity by regulating a series of signal pathways, correcting the imbalance of remodeling [11]. 

Graphene oxide (GO), prepared by oxidation of graphite, is a two-dimensional carbon-based nanomaterial with many hydrophilic functional groups (i.e., hydroxyl, carboxyl and epoxy groups), favorable biocompatibility and physiochemical stability compared with pristine graphene [12]. However, bulk graphene-based porous structures have small pore sizes and lack enough mechanical strengths, leading to low cell adhesion and slow migration on these surfaces [13,14]. Also, cytotoxicity is related to the size, shape and concentration of GO nanoparticles, which need to be carefully considered [15,16]. It is noteworthy that GO can be readily functionalized with other materials because of its large specific surface area, as well as hydrophilic groups, π–π stacking framework and electrostatic interactions [17,18].

In recent years, the use of calcium phosphate ceramics (CaPs) is of increasing interest as a strategy to design the inorganic part of biomimetic scaffolds. Among the CaPs, β-tricalcium phosphate (β-TCP) has good biocompatibility, osteoconductivity, and resorbability, which can form a fast fixation and chemical connection with bone [19,20,21,22]. 

Herein, we hypothesize that incorporating β-TCP and GO nanoparticles into the SF/SPI scaffolds could invariably improve the physicochemical and biological properties, particularly, enhance osteoinductive capability of the SF/SPI scaffolds. The objective of our study is to investigate the synergistic effect on promoting the osteogenic differentiation of bone marrow mesenchymal stem cells (BMSCs) in the presence of β-TCP and GO nanoparticles. A series of SF/SPI-based composite scaffolds for bone tissue engineering are fabricated by incorporating β-TCP and GO nanoparticles simultaneously or individually. The physical, chemical and biological properties of these SF/SPI-based composite scaffolds are evaluated.

## 2. Materials and Methods

### 2.1. Preparation of Silk Fibroin Solution

Aqueous SF solution was prepared based on a previously reported procedure [23]. Briefly, 8 g Bombyx mori cocoon was finely cut into pieces, and boiled in an aqueous solution of 0.02 M Na_2_CO_3_ for 60 min. After being rinsed with deionized water three times, the samples were dehydrated into an oven at 60 °C overnight. The dried silk fibroin was sequentially dissolved in 100 mL 9.3 M LiBr solution, and kept stirring until complete dissolution at 60 °C. Then, the solution was dialyzed in deionized water, using a dialysis tube with molecular weight cutoff of 10,000 Da for three days. The dialyzed SF solution was centrifuged at 6000 r/min for 10 min to remove the insoluble part, and was stored at 4 °C.

### 2.2. Fabrication of SF/SPI-Based Composite Scaffolds

SF/SPI-based composite scaffolds containing SF, SPI, β-TCP, and GO were prepared by a freeze-drying method. First, SF and SPI were prepared as 4% (*w*/*v*) stock solutions in distilled water respectively. The SF and SPI solutions were mixed in the volume ratio of 1:1, and was kept stirring at room temperature until there were no bubbles. Then, β-TCP and GO particles were sequentially added into the mixed SF/SPI solution in the ratio of 10 mg:10 mg:10 mL, and transferred to a 48-well plate and frozen at −80 °C for 24 h followed by lyophilization. The lyophilized scaffolds were immersed in the 4-morpholine ethane sulfonic acid (MES) buffer solution (50 mmol/L, 70% ethanol solution) containing EDC/NHS (EDC 50 mmol/L, NHS 25 mol/L) for 24 h at room temperature as crosslinking step [24,25]. The crosslinked scaffolds were immersed into 75% ethanol and deionized water to rinse the redundant crosslinking reagent, and frozen at −80 °C for 24 h, followed by vacuum drying for 36 h (SCIENTZ-10N, SCIENTZ, Ningbo, China). At last, the SF/SPI/GO/β-TCP composite scaffolds were stored in a container for future use. To fabricate SF/SPI, SF/SPI/β-TCP and SF/SPI/GO scaffolds, the above protocol was followed in where the addition of β-TCP or GO was excluded.

### 2.3. Characterization of the SF/SPI-Based Composite Scaffolds

#### 2.3.1. Morphologies of the SF/SPI-Based Composite Scaffolds

Surface morphologies of the prepared samples were observed by a field emission scanning electron microscope (FESEM) (XL-30, Philips, NLD, Amsterdam, The Netherlands). The samples were first sputter-coated with gold using ion sputter. FESEM photographs were obtained at magnifications of 500× and 5000× with an acceleration voltage of 15 kV.

#### 2.3.2. Compositions of the SF/SPI-Based Composite Scaffolds

Chemical composition of the prepared samples were analyzed via a X-ray diffraction (XRD) (X′ Pert^3^ Powder, PANalytical, NLD, Amsterdam, The Netherlands) and attenuated total reflectance Fourier transformed infrared spectroscopy (ATR-FTIR) (VERTEX 70, Bruker, GER, Billerica, MA, USA). XRD analysis was performed using a Cu-Kα source, and the spectrum were recorded from 8 to 80° 2 theta at 36 kV. All characteristic peaks were identified in the International Centre for Diffraction Data (ICDD) database. ATR-FTIR was used to analyze the chemical compositions of different samples. The spectra were recorded from 4000–400 cm^−1^ with a 2 cm^−1^ resolution and 20 scans.

#### 2.3.3. Pore Size and Porosity of the SF/SPI-Based Composite Scaffolds

Porosity of all scaffolds was measured using a previously described liquid displacement method [26]. All samples were first cut into the same size, and completely immersed into measuring cylinder with a known volume of anhydrous ethanol (V1). The volume after the immersion of all sample scaffolds was recorded (V2). Finally, the ethanol-impregnated scaffolds were removed from the measuring cylinder. The residual ethanol volume was recorded (V3). The porosity (P) of the scaffold was calculated through the following equation. Each test was performed in three independent preparations. P = [(V1 − V3)/(V2 − V3)] × 100%(1)

#### 2.3.4. Water Adsorption of the SF/SPI-Based Composite Scaffolds

Water adsorption behaviors (swelling ratios) of the SF/SPI-based composite scaffolds were measured using a gravimetric method. All sample scaffolds were cut into the same size. The weight of dry sample scaffolds was recorded (M0). After immersion into PBS solution for 24 h, the excess water was removed using a filter paper. The sample scaffolds were then dried in an oven at 60 °C under vacuum overnight, and the dry weight of the scaffolds (M1) was recorded. Water adsorption ratio (%) = [(M0 − M1)/M0] × 100%(2)

#### 2.3.5. In Vitro Mineralization of the SF/SPI-Based Composite Scaffolds

In vitro mineralization was performed by immersing the SF/SPI-based composite scaffolds into simulated body fluid (SBF). The SBF solution was prepared as reported in a previous work [27]. All the prepared samples (1 × 1 × 1 cm^3^) were immersed in the 5 mL SBF solution and incubated at 37 °C for 1, 3, 7, and 14 days. At the planned time point, the samples were rinsed with deionized water and subsequently lyophilized. The formed apatite on surface of all SF/SPI-based scaffolds was observed using scanning electron microscope (SEM) (XL-30, Philips, NLD, Amsterdam, The Netherlands) after being immersed into the SBF solution at different time points, while the formed apatite composition was determined using XRD with the scanning angles between 20° and 70° on 14th day.

#### 2.3.6. Mechanical Properties of the SF/SPI-Based Composite Scaffolds

Mechanical properties of the SF/SPI-based composite scaffolds were calculated using a Universal Testing Machine (Instron 5967, Instron LTD, Boston, MA, USA). A 0.5 force was loaded on sample scaffolds with a measuring speed of 20 mm/min. Three tests for compressive strength were performed in each sample.

### 2.4. In Vitro Biocompatibility and Osteogenic Evaluation of the SF/SPI-Based Composite Scaffolds

#### 2.4.1. Morphology and Viability of BMMCs on the SF/SPI-Based Composite Scaffolds

Sprague-Dawley (SD) rat bone marrow mesenchymal stem cells (BMSCs) (Cyagen Biociences, Guangzhou, China) were cultured in BMSCs growth medium (Cyagen Biociences, Guangzhou, China) containing 10% fetal bovine serum (FBS) and 1% antibiotic-antimycotic at 37 °C with 5% CO_2_. The culture medium was changed every two days until the cells reached 80–90% confluence. The third cell passage was used for proliferative evaluation, and the fifth cell passage was used for differentiation.

Cells were cultured at a density of 2 × 10^4^ per sample on the SF/SPI-based composite scaffolds for 1, 3, 5, and 7 days. Cell morphology on each scaffold was observed at 48 h using SEM and confocal microscopy. Cell viability was evaluated using Cell Counting Kit-8 (CCK-8, Beyotime, Shanghai, China). The scaffolds of SF/SPI/GO and SF/SPI/GO/β-TCP without cell seeding are used as controls. At each time point, the samples were washed twice with PBS. Fresh culture medium (200 μL) was mixed with 20 μL of CCK-8 reagent and added to each sample. Then, the culture was incubated at 37 °C for 2 h. After the incubation, 100 μL of the medium was transferred to a 96-well plate and measured at 450 nm.

#### 2.4.2. Cell Cytoskeletal Organization of BMSCs on the SF/SPI-Based Composite Scaffolds

BMSCs were seeded on the SF/SPI-based composite scaffolds at a density of 4 × 10^5^ cells per sample. For actin staining, cells were washed gently with PBS, and fixed with 10% neutral buffered formalin for 15 min at room temperature. The cells were permeabilized using 0.5% Triton X-100 in PBS for 5 min. After being washed with PBS for 10 min twice, the cells were incubated with rhodamine-conjugated phalloidin (CA1610, Solarbio, Beijing, China) for 30 min in the dark followed by counterstaining with 4′,6-diamidino-2-phenylindole (DAPI) to visualize the nuclei. Images were captured at 40× magnification with a confocal microscope (Nikon AIR, Nikon, Tokyo, Japan).

#### 2.4.3. Alkaline Phosphatase Activity of BMSCs on the SF/SPI-Based Composite Scaffolds

ALP activity is regarded as a marker of early stage osteogenic expression. Cells were seeded at a density of 2 × 10^4^ per sample on the SF/SPI-based composite scaffold for 3, 5, or 14 days. ALP activity of the BMSCs was measured using ALP assay kit (Beyotime, Shanghai, China) which was based on the color reaction of colorless *p*-nitrophenyl phosphate (pNPP) converted to yellow p-nitrophenol after incubation at 37 °C for 30 min. Briefly, the original culture medium was removed, and the cells were washed twice with PBS at each time point, and sequentially lysed using RIPA lysis buffer. After centrifugation at 5000 rpm for 10 min, the supernatant was transferred into 96-well plate. The substrates and *p*-nitrophenol were added in sequence and incubated at 37 °C for 30 min. Finally, the reaction was stopped by the addition of 100 μL stop buffer and the absorbance at 405 nm was measured.

#### 2.4.4. Osteogenesis-Related Gene Expression of BMSCs on the SF/SPI-Based Composite Scaffolds

Runt-related transcription factor 2 (Runx 2), osteocalcin (OC), and collagen type I (Col I) are representative osteogenesis-related gene markers. Quantitative real-time reverse transcription polymerase chain reaction (qRT-PCR) was performed to evaluate the osteogenesis-related gene expression of BMMCs cultured on the SF/SPI-based scaffolds on day 7 and 14. At each time point, the total RNA from each sample was first extracted using TRIzol solution (Life Technologies, Carlsbad, CA, USA), and the concentration was determined with the spectrophotometer (NanoDrop 22 Technologies, Shanghai, China). The primer sequences used for PCR amplification are listed in Table 1. The gene expression levels were calculated using the 2^−ΔΔCT^ method.

### 2.5. Statistics Analysis

Statistical analysis was performed using SPSS 15.0 software. All data are depicted as mean ± standard deviation values. Multi-factor analysis of variance was first used to evaluate the statistical significance among five groups. Paired t test was used to evaluate the differences between each of the two groups at four or eight weeks. A *p* value of less than 0.05 was considered as statistically significant.

## 3. Results

### 3.1. Physical and Chemical Properties of the SF/SPI-Based Composite Scaffolds

#### 3.1.1. Morphologies of the SF/SPI-Based Composite Scaffolds

Generally, all scaffolds display cylindrical porous structures, with a diameter of 9 mm and thickness of 6 mm fabricated through the freeze-drying method (Figure 1). The microstructures of the SF/SPI-based composite scaffolds were characterized with SEM as shown in Figure 2. All scaffolds represent similar microporous morphology and good connectivity between the pores. Most of the SF and SF/SPI scaffolds exhibit ellipse-shaped pores with rounded pore edges (Figure 2A,B). Obviously, the scaffolds display a flake-like structure with sharp pore edges when β-TCP is incorporated (Figure 2D,E). The shapes of the pore edges are altered when the β-TCP or GO nanoparticles appear on the surface of the hole walls (Figure 2C–E). These changes can be detected from the magnified images on the top right corner of each figures (Figure 2A–E).

The pore sizes and porosities of the composite scaffolds are summarized in Table 2. The SF scaffold has the smallest pore size (117.76 ± 6.33 μm), while no significant difference is found on the scaffolds after loading of SPI. The pore sizes of the SF/SPI and SF/SPI/GO scaffolds are 113.37 ± 6.33 and 108 ± 6.33 μm, respectively. The pore sizes of the SF/SPI/β-TCP and SF/SFI/β-TCP/GO scaffolds increase in the presence of β-TCP, which are significantly larger than those of the other scaffolds (232.53 ± 4.09 and 194 ± 6.18 μm). All the composite scaffolds present similar porosities, which have no significant difference between each other. The SF/SPI/GO scaffold has the highest porosity (87.66 ± 2.77)%, while the control SF scaffold has the lowest porosity (79.32 ± 1.62)%. The porosities of the SF/SPI, SF/SPI/β-TCP, and SF/SPI/GO/β-TCP scaffolds are (82.28 ± 2.15)%, (82.63 ± 1.04)%, and (80.45 ± 2.04)%, respectively.

#### 3.1.2. Mechanical Properties of the SF/SPI-Based Composite Scaffolds

Mechanical properties of the composite scaffolds are shown in Figure 3. The SF scaffold had the lowest compressive strength (0.333 ± 0.085 MPa). The compressive strengths of the SF/SPI and SF/SPI/GO scaffolds were 0.432 ± 0.072 and 0.660 ± 0.022 MPa, respectively. The compressive strengths of the SF/SPI/β-TCP and SF/SPI/GO/β-TCP scaffolds were 1.020 ± 0.122 MPa and 0.802 ± 0.065 MPa respectively, which were significantly higher than those of the SF-based scaffolds without β-TCP particles (*p* < 0.05).

#### 3.1.3. Chemical Constituents of the SF/SPI-Based Composite Scaffolds

Figure 4 shows the XRD pattern of the SF/SPI-based composite scaffolds. A weak peak locates at 9.7° and 20.2° corresponding to the β-sheet crystalline structure (silk-II structure) of native SF. However, the weak signal of SF around 20° is overlapped by the signal of SPI in the hybrid scaffolds, which is the characteristic diffraction peaks at 19.7° in the SPI XRD pattern corresponding to the β-sheet structures of the protein secondary conformation. When β-TCP nanoparticles are incorporated, the peaks occur at 31.0° and 34.3°, and match the standard β-TCP JCPDS 09-0169 card. The characteristic peak (001) of GO at 10.2° is not observed in GO-incorporated scaffolds in the present study, probably because of the low content or the overlap position of the SF characteristic peak at 9.7°.

ATR-FTIR patterns of the SF/SPI-based composite scaffolds are shown in Figure 5. All of the scaffolds exhibit characteristic peaks at 1612 cm^−1^ (C=O stretching vibration), 1513 cm^−1^ (N-H bending), and 1298 cm^−1^ (C-H and N-H stretching), which could be ascribed to amide I, amide II, and amide III of SF respectively [28,29]. Also, the ATR-FTIR spectrum of SPI displays similar characteristic peaks (1630 cm^−1^, 1530 cm^−1^, and 1230 cm^−1^) to that of SF based on several literatures, which are difficult to distinguish in hybrid scaffolds [30]. Hydroxy groups (-OHs) of SF appear at 3260 cm^−1^, the N-H stretching vibration of SPI is at 3460 cm^−1^. When SF is mixed with SPI, the hybrid characteristic peak becomes obviously broad, and moves to lower adsorption band (3400 cm^−1^), indicating that the two proteins are not only physically blended together, but interacted though hydrogen bonding. The SF/SPI/β-TCP and SF/SPI/GO/β-TCP scaffolds exhibit the characteristic peak at 660 cm^−1^, which attributes to PO_4_^−3^ groups [29,31,32]. The addition of GO enhances the intensity of -OH groups (3200 cm^−1^ to 3700 cm^−1^) with broader band, representing the intermolecular hydrogen bonding in the scaffold. The absorption peaks at 1732 cm^−1^ and 1628 cm^−1^ are ascribed to the C=O stretching vibration of carboxyl groups and C=C stretching of sp2 hybridized crystal structures of graphite, respectively [33]. Only the adsorption peak of GO at 1732 cm^−1^ is detected in the composite scaffolds, while the adsorption peak at 1628 cm^−1^ is overlapped by the adsorption band of amide I.

Water adsorption behaviors of the SF/SPI composite scaffolds are shown in Figure 6. All of the scaffolds display good water adsorption behaviors in 24 h. The control SF scaffold rapidly reaches the swelling ratio of 1465% at 3 h, and tends to be constant, i.e., 1661% at 24 h. The scaffolds SPI, SF/SPI exhibit a rapid increase up to 2346% after 3 h immersion, and become stable (2321%) until 24 h. The SF/SPI/GO scaffold has the highest swelling ratio (i.e., 2668%) at 4 h, slightly decreases to 2540% at 12 h, and finally reaches 2560% at 24 h. The water adsorption ratios of the SF/SPI/β-TCP and SF/SPI/GO/β-TCP scaffolds initially have a slow increase, when compared with pure SF scaffold. After 4 h immersion, the SF/SPI/β-TCP and SF/SPI/GO/β-TCP scaffolds exhibit a rapid increase up to 1508% and 1569%, and approximately keep 1450% and 1856% at 24 h, respectively. The water adsorption ratios of the SF/SPI and SF/SPI/GO scaffolds are significantly higher than those of the SF and SF/SPI/β-TCP scaffolds at 24 h (*p* < 0.05).

### 3.2. In Vitro Biomineralization Capability of the SF/SPI-Based Composite Scaffolds

Biomineralization capability is an essential factor to promote bone-binding competence for bone repair materials. The surface morphologies of the Ca-P minerals on the composite scaffolds are confirmed by the SEM images (Figure 7) and XRD analyses (Figure 8). Generally, no mineral deposition is found on each scaffold until the 3rd day. After incubation in the SBF solution for 5 days, lamellar nanocrystals are observed on the surface of each scaffold. Obviously, there are more mineral deposition on the surface of each scaffold with the increased immersion time. It is interesting that the minerals are sparsely and randomly distributed on the SF scaffold, while there are denser mineral aggregations on the scaffolds with the presence of GO or β-TCP nanoparticles. Besides, the SF/SPI/GO scaffold displays more lamellar nanocrystals, when compared with SF and SF/SPI scaffolds, indicating the nanocrystal deposition is increased in the presence of GO. With the incorporation of β-TCP nanoparticles, there are more nanocrystals formed on the surface of the SF/SPI/β-TCP scaffold. However, no differences are observed between the SF/SPI/β-TCP and SF/SPI/GO/β-/TCP scaffolds via SEM.

As shown in Figure 8, the XRD patterns after incubation in SBF solution for 14 days display the characteristic peaks corresponding to apatite at 26.0° (002) and 31.77° (211) (JCPDS 9432 card), which confirm the nucleation of hydroxyapatite (HA) in all the samples. Extremely weak peak at 26.0° is observed on SF scaffold, indicating small amount and low crystallinity of apatite formed. The SF/SPI scaffold demonstrates a similar extent of mineralization as that of the SF/SPI/GO. Additionally, the narrow peaks at 26° and 31° with strongest intensities, as well as the particular peak at 54° on SF/SPI/GO/β-TCP scaffold illustrate that there are more HA particles with higher crystallinity and larger size deposited on the SF/SPI/GO/β-TCP scaffold.

### 3.3. Morphologies and Proliferation of BMSCs on the SF/SPI-Based Composite Scaffolds

Cell morphologies of BMSCs on the SF/SPI-based composite scaffolds are observed at 48 h using both SEM and confocal microscopy. As shown in Figure 9, there are elongated spindle-sharped cells well adhered onto the surface of the SF/SPI/GO, SF/SFI/β-TCP, and SF/SPI/GO/β-TCP scaffolds, while there are less cells on the SF and SF/SPI scaffolds. Obviously, cells on the SF/SPI/GO/β-TCP scaffold exhibit the most spreading and adhesive morphologies.

BMSC proliferation rate on the composite scaffolds is investigated using CCK-8 kit on 1, 3, 5, and 7 days. As shown in Figure 10, cell viability on the SF scaffold decreases slightly with time increasing, which remains approximately 80%, while cell viabilities increase on the other four scaffolds with increasing time. This may be due to the reason that there are not sufficient receptors on the surface of the SF scaffold for cell adhesion and spreading. Cell viabilities on the SF/SPI and SF/SPI/GO scaffolds remained the same as that of the 1st day, which was significantly higher than those of the other three scaffolds (*p* < 0.05). No differences were found on each scaffold on the 3rd day. Since the 5th day, cell viabilities on the SF and SF/SPI/β-TCP scaffolds slightly decrease. Meanwhile, cell viabilities on the SF/SPI/GO and SF/SPI/GO/β-TCP scaffolds exhibit the highest levels, which are significantly higher than the other three groups (*p* < 0.05). On the 7th day, the SF/SPI/GO and SF/SPI/GO/β-TCP scaffolds still exhibit the highest cell viabilities (*p* < 0.05).

These results are consistent with the confocal images as shown in Figure 11, where BMSCs behave totally different on the SF/SPI-based scaffolds. On the pure SF and SF/SPI scaffolds the nuclei are blue and clear but the cytoplasm and cytoskeleton are sparse and vague (Figure 11A,B). On the SF/SPI/GO scaffold, the numbers of cells are obviously increased with many divisions (i.e., double nuclei, Figure 11C). Cell-cell interactions are extremely active with stretched pseudopodia. On the SF/SPI/β-TCP scaffold, some cells are at division states with spread cytoskeleton (Figure 11D). While on the SF/SPI/GO/β-TCP scaffold, cell activities attend the highest with bright blue nuclei and fibrous orange cytoskeleton (Figure 11E). There are more elongated and spindle cell shapes on the composite scaffolds with β-TCP or GO nanoparticles, when compared with the pure SF and SF/SPI scaffolds.

### 3.4. ALP Activities of BMSCs on the SF/SPI-Based Composite Scaffolds

ALP is widely used as a maker of early differentiation of MSCs. ALP activities of BMSCs on different scaffolds on day 1, 3, 5, and 7 are shown in Figure 12. On the 1st day, the SF/SPI/β-TCP group has a significantly higher ALP level than the other four scaffolds (*p* < 0.05). No significant differences are found on the ALP activity between the SF, SF/SPI, SF/SPI/GO, and SF/SPI/GO/β-TCP groups. ALP activities on all scaffolds except the SF/SPI/GO group slightly decrease on the 3rd day. ALP activities almost keep the same for the SF/SPI/GO and SF/SPI/β-TCP groups, which are significantly higher than those of the other three groups (*p* < 0.05). ALP activities on all scaffolds rapidly increase on the 5th day and decrease on the 7th day. On the 5th day, the SF/SPI/GO/β-TCP has the highest ALP activity, which is almost two-fold higher than that of the SF/SPI/GO group (*p* < 0.05). The SF/SPI/GO group has a significantly higher ALP level than the SF, SF/SPI, and SF/SPI/β-TCP groups (*p* < 0.05), while no differences are found on ALP level between the SF/SPI and SF/SPI/β-TCP groups. On the 7th day, ALP activities of all the groups almost keep at the same level without any differences.

### 3.5. Osteogenesis-Related Gene Expression of BMSCs on the SF/SPI-Based Composite Scaffolds

Runx 2 is known as an osteogenic transcriptional factor, as well as an early marker of osteogenic differentiation. Figure 13 displays the Runx 2 mRNA level of BMSCs cultured on the SF/SPI-based composite scaffolds. On the 7th day, the SF/SPI/GO scaffold has the highest Runx2 level, which is significantly higher than those of the SF and SF/SPI/β-TCP scaffolds (*p* < 0.05). Runx 2 level on the SF/SPI/β-TCP/GO scaffold is slightly lower than that of the SF/SPI/GO scaffold, and significantly higher than that of the SF scaffold (*p* < 0.05). After 14 days of culture, the Runx 2 level slightly decreases on the SF and SF/SPI scaffolds. Interestingly, Runx 2 level on the SF/SPI/GO/β-TCP scaffold is up-regulated the most, which is significantly higher than those of the SF, SF/SPI, and SF/SPI/β-TCP scaffolds (*p* < 0.05). Additionally, the SF/SPI/GO group has a significantly higher Runx 2 level, when compared with the SF and SF/SPI groups (*p* < 0.05).

Osteocalcin (OC), a specific protein related to osteoblasts, has been widely used as a late marker of osteogenic differentiation. As shown in Figure 14, all scaffolds have an increasing OC level with time increasing. The SF/SPI/GO/β-TCP group exhibits the highest OC level on the 7th day, which is significantly higher than that of the SF group (*p* < 0.05). OC level of BMSCs on the SF/SPI/GO group is significantly higher than that of the SF and SF/SPI/β-TCP groups on the 14th day (*p* < 0.05). No differences are found on OC level between the SF, SF/SPI, and SF/SPI/GO groups.

Collagen type I (Col I) is the main component in the composition of ECMs. It is an important marker protein in osteogenesis. As shown in Figure 15, the SF/SPI, SF/SPI/β-TCP, and SF/SPI/GO/β-TCP groups have an increasing Col I level with time increasing, while Col I level on the SF and SF/SPI/GO groups decreases slightly from the day 7 to 14. On the 7th day, there are significantly increased OC levels on the SF/SPI/GO/β-TCP and SF/SPI/GO scaffolds compared with the SF scaffold (*p* < 0.05). In addition, the Col I level reaches the highest value on the SF/SPI/GO/β-TCP scaffold on the 14th day, that is considerably higher than those of the other four scaffolds (*p* < 0.05). OC levels of BMSCs on the SF/SPI/GO and SF/SPI/β-TCP scaffolds are significantly higher than those of the SF and SF/SPI scaffolds (*p* < 0.05).

## 4. Discussion

SF and SPI are both natural polymers that have been widely used as scaffolds for tissue engineering and regenerative medicine [34,35,36,37]. However, there are many limitations for these polymers to be used as bone repair scaffolds because of the poor mechanical properties and insufficient osteoinductive capabilities.

It is generally accepted that the pore geometry in a scaffold, including pore size, pore shape, porosity, and pore interconnecting pattern, plays an important role in cell adhesion, proliferation, and migration as well as tissue ingrowth [38,39]. Lager pore sizes favor tissue ingrowth and vascularization, resulting in better osteogenesis, while smaller pores lead to osteochondral ossification [40]. An ideal bone scaffold should have 100–600 μm pore size with 60–80% porosity to provide enough space for cell migration, tissue ingrowth, and vascularization.

In this study, all the SF/SPI scaffolds fabricated through freeze-drying have an average pore size of more than 100 μm and proper porosities. With these pores, the amount of the incorporated β-TCP and GO nanoparticles can be significantly increased, which subsequently affects the hydrophobicity of the whole scaffolds [29,31]. When the β-TCP and GO nanoparticles appear on the surface of the pore walls, there form flake-like structures with sharp pore edges, which increase the surface roughness. The good dispersion of the nanoparticles of β-TCP and GO and the altered surface properties could be helpful to enhance cell behaviors [36,41,42].

It is considered that appropriate mechanical properties are necessary for bone tissue engineering. In the present study, the single SF scaffold has low mechanical properties (0.33 MPa), while the SF/SPI composite scaffold holds significantly increased compressive strength (0.42 MPa). This could be explained by the effect of chemical-crosslinked polymer chains. Furthermore, the reinforcement of GO increases the compressive strength value of the scaffold to 0.66 MPa. It could be due to the formation of hydrogen bonds among -OH and -COOH functional groups on the GO surface and hydroxyl groups in the SF or SPI molecules [18,43]. The incorporation of β-TCP particles can obviously enhance the mechanical properties of the SF/SPI scaffolds. It is noted that the compressive strength (i.e., 1.020 ± 0.122 MPa) of the SF/SPI/β-TCP scaffold is similar to that of the natural trabecular bone (1–7 MPa) [44]. This is beneficial for bone repair without the help of extra mechanical support [45,46,47,48,49].

It is generally accepted that in vitro biomineralization of a porous scaffold through a layer of an apatite formation over the scaffold surface could provide direct integration between host bone and the bioactive mineral layers, thereby accelerating bone healing [27]. The SBF solution is often used to imitate a physiological environment, which can nucleate bone-like HA on bone repair materials [50]. The HA particles formed on the scaffolds can be increased with the increasing incubation time. Previous studies demonstrated the biomineralization process was influenced by the functional groups (i.e., amino, carboxyl and hydroxyl group), surface charge, and surface morphology [51,52].

In the present study, both SF and SF/SPI scaffolds lead to a similar HA nucleation, which could be explained by the fact that both SF and SPI molecules have similar amino acid compositions. Interestingly, with the incorporation of GO with enriched functional groups, no differences are found on the intensity and sharp degrees of the XRD peaks of the SF/SPI and SF/SPI/GO scaffolds (Figure 5), indicating the types of functional groups are not the decisive factor that affected the biomineralization processes [24,53,54,55,56,57]. However, more characteristic peaks with considerably greater intensities are found in the presence of β-TCP, indicating a greater extent of mineralization on the SF/SPI/β-TCP and SF/SPI/GO/β-TCP scaffolds. It is speculated that more calcium ions (Ca^2+^) released from the β-TCP nanoparticles could incorporate with the functional groups through electrostatic interactions, subsequently formed minerals with phosphate groups (PO_4_^−3^), and facilitated the nucleation and growth of HA crystals on the surface of the scaffolds [58]. Additionally, the rough flake-like morphologies on the β-TCP-incorporated scaffolds are also attributed to the enhanced HA nucleation [59]. Thus, along with the larger specific area of GO, the SF/SPI/GO/β-TCP scaffold exhibits the most biomineralization capability in this study.

Some researchers suggested that the initial cell adhesion and proliferation states at material-tissue interface play a crucial role in the early stage of bone formation. In the present study, BMSCs are well adhered on all the scaffolds though SEM and confocal observation. Compared to pure SF and SF/SPI scaffolds, the incorporation of β-TCP and GO nanoparticles enhances the cell adhesive capability. Along with CCK-8 results, the simultaneous incorporation of β-TCP and GO nanoparticles into SF/SPI scaffolds has a better cell morphology and viability compared to GO or β-TCP alone. GO nanoparticles provide larger specific surface area, which facilitate more cell adherence onto the scaffold surface with well-spread morphologies. Cell behaviors can also be influenced by the hydrophilicity of the scaffold surface [60,61,62,63,64]. The hydrophilic property of GO-incorporated SF/SPI scaffolds is likely responsible for the increasing swell ratio. Therefore, the enhancement of water adsorption behaviors on GO-incorporated composite scaffolds facilitates the adhesion and proliferation of the BMSCs [65,66,67,68,69].

The osteoinductive ability of composite scaffolds are investigated using ALP activities, through measuring the Runx 2, OC and Col I expression levels of BMSCs. It is found that the incorporation of β-TCP and GO nanoparticles into the composite scaffolds has significantly increased the mRNA levels of osteogenesis-related genes, indicating the synergistic effect in the osteogenesis of BMSCs via incorporating β-TCP and GO nanoparticles. It was reported that the surface characteristics of GO could influence the molecular pathway of stem cells [70]. Nevertheless, the regulatory mechanism of β-TCP and GO nanoparticles on osteogenesis remains poorly understood. In the present study, most of the GO flakes are covered by β-TCP particles, and formed GO-β-TCP complexes on the pore edges, which are beneficial for the direct absorption of molecules. The enhanced osteogenic differentiation capability might be partly due to the increased interaction between the intracellular focal adhesion complexes and GO-β-TCP structures [71]. Partly, the incorporation of β-TCP has increased the mechanical properties, which could induce an improved mechanotransduction effect to regulate cells differentiation. Actually, our results are in accordance with previous studies, which indicate that novel GO-CaP nanocomposites could synthetically promote osteogenesis of MSCs and further enhance calcium deposition by osteoblasts [72,73]. As discussed above, the simultaneous incorporation of β-TCP and GO nanoparticles facilitates biomineralization, and subsequentially provides a biomimetic environment for BMSC osteogenesis. Further elucidation of the underlying molecular mechanism and signal pathway related BMSCs osteogenesis through β-TCP and GO nanoparticles will be conducted in the next study.

## 5. Conclusions

In present study, a series of SF/SPI-based composite scaffolds are fabricated by incorporating β-TCP and GO nanoparticles individually or simultaneously. The incorporation of β-TCP has significantly increased the pore size and compressive strength, while the incorporation of GO has enhanced water adsorption behaviors of the composite scaffolds. More specifically, simultaneous incorporation of β-TCP and GO particles has facilitated biomineralization with an obvious synergistic effect on improving the adhesion, proliferation, and osteogenic differentiation of the BMSCs. In another word, the cooperation of GO and β-TCP nanoparticles possesses synergistic effect in osteogenesis of BMSCs compared with GO or β-TCP nanoparticle alone. It is expected that the GO and β-TCP nanoparticle incorporated SF/SPI scaffolds will be good candidates for future bone tissue regeneration.

## Figures and Tables

**Figure 1 polymers-12-00069-f001:**
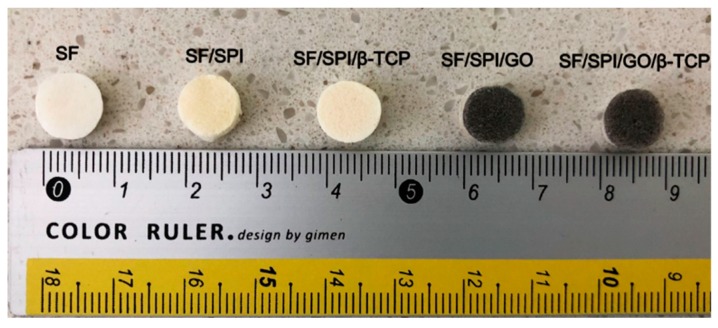
General observation of the SF/SPI-based composite scaffolds.

**Figure 2 polymers-12-00069-f002:**
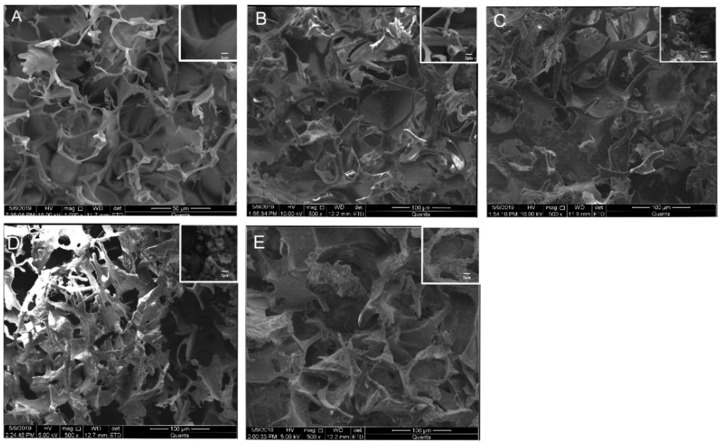
Surface morphology of the silk fibroin/soy protein isolate (SF/SPI)-based composite scaffolds ((**A**): SF; (**B**): SF/SPI; (**C**): SF/SPI/graphene oxide (GO); (**D**): SF/SPI/β-tricalcium phosphate (β-TCP); (**E**): SF/SPI/GO/β-TCP). Enlarged views of the pore structures are on the top right corner of each figures.

**Figure 3 polymers-12-00069-f003:**
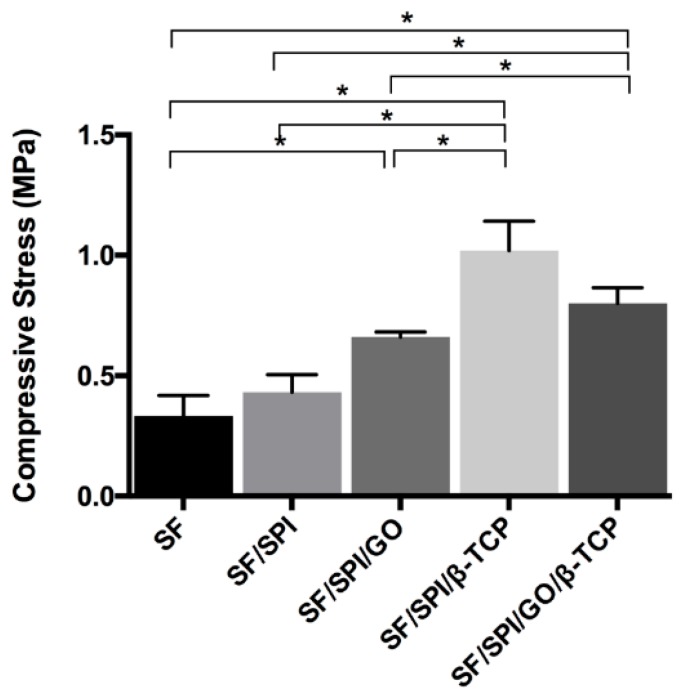
Mechanical properties of the SF/SPI-based composite scaffolds. Statistical significance relative to each group: * *p* < 0.05.

**Figure 4 polymers-12-00069-f004:**
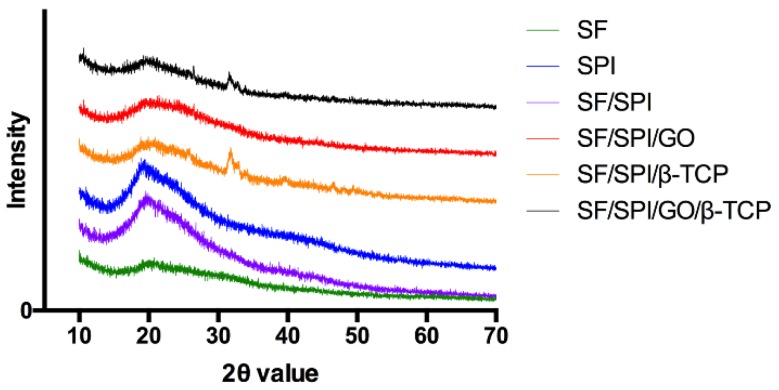
X-ray diffraction (XRD) patterns of the SF/SPI-based composite scaffolds.

**Figure 5 polymers-12-00069-f005:**
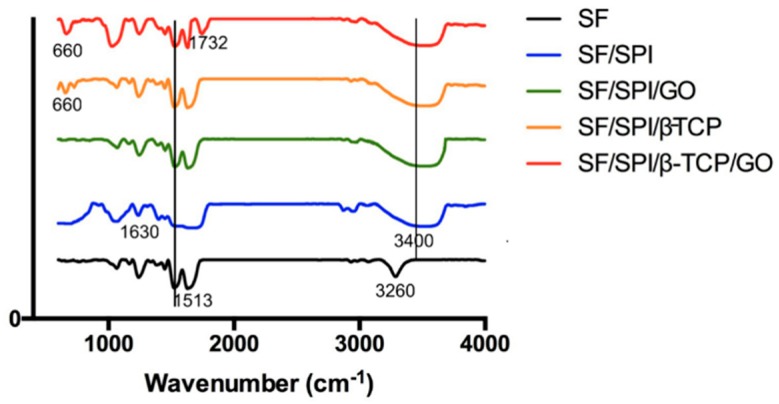
ATR-FTIR patterns of the SF/SPI-based composite scaffolds.

**Figure 6 polymers-12-00069-f006:**
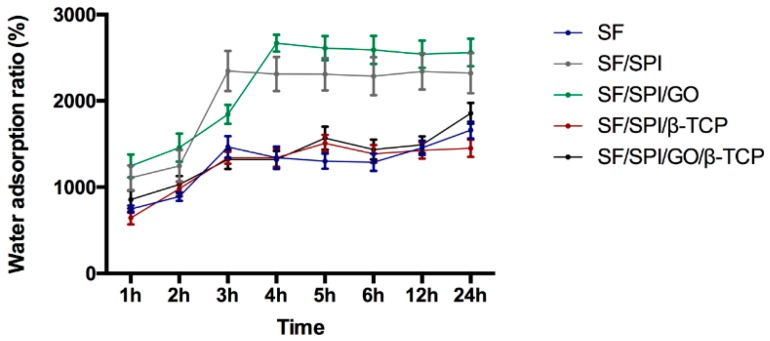
Water adsorption behaviors of the SF/SPI-based composite scaffolds in 24 h.

**Figure 7 polymers-12-00069-f007:**
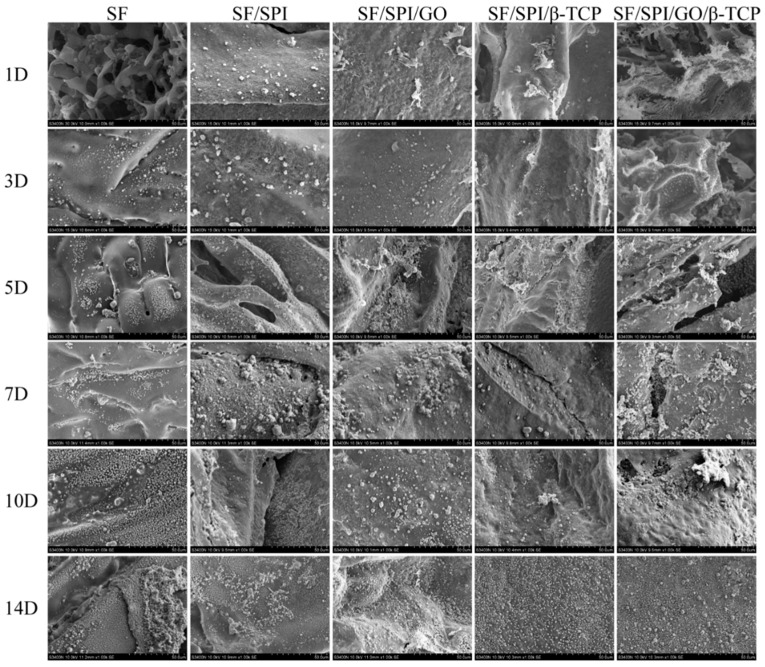
Field emission scanning electron microscope (FESEM) photographs of the SF/SPI-based composite scaffolds after immersing into simulated body fluid (SBF) solution at different time points.

**Figure 8 polymers-12-00069-f008:**
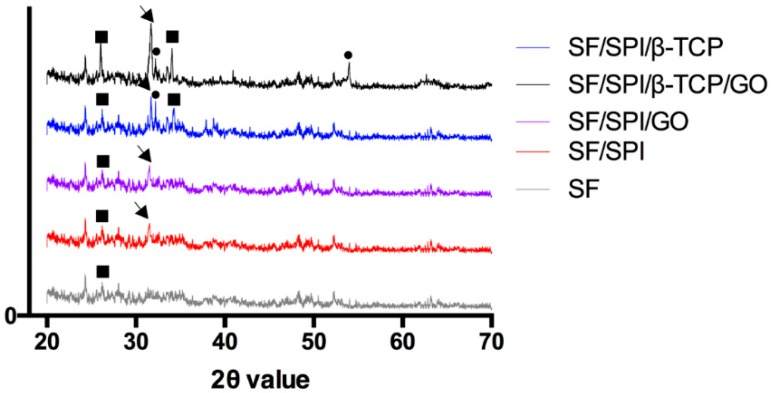
XRD patterns of the SF/SPI-based composite scaffolds after immersing into SBF solution for 14 days (circle refers to hydroxyapatite; arrow refers to apatite; square refers to β-TCP).

**Figure 9 polymers-12-00069-f009:**
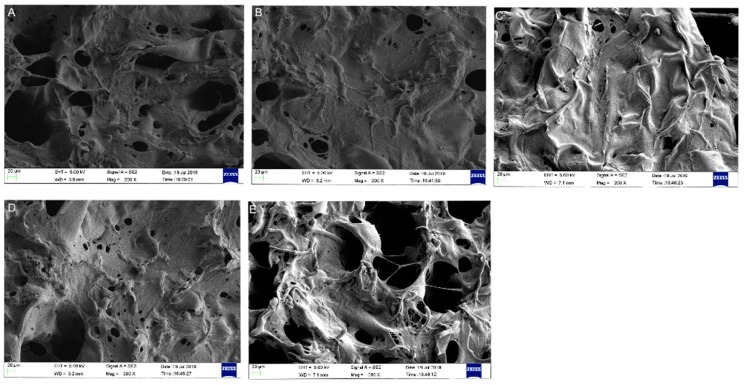
Scanning electron microscope (SEM) photographs of bone marrow mesenchymal stem cells (BMSCs) seeding on the SF/SPI-based composite scaffolds ((**A**): SF; (**B**): SF/SPI; (**C**): SF/SPI/GO; (**D**): SF/SPI/β-TCP; (**E**): SF/SPI/GO/β-TCP).

**Figure 10 polymers-12-00069-f010:**
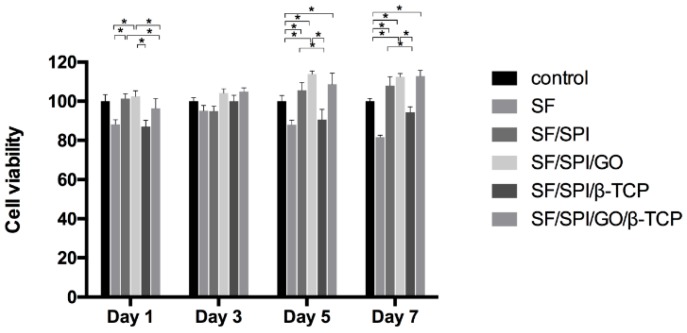
Proliferation of BMSCs on the SF/SPI-based scaffolds on day 1, 3, 5, and 7. Statistical significance relative to each group: * *p* < 0.05.

**Figure 11 polymers-12-00069-f011:**
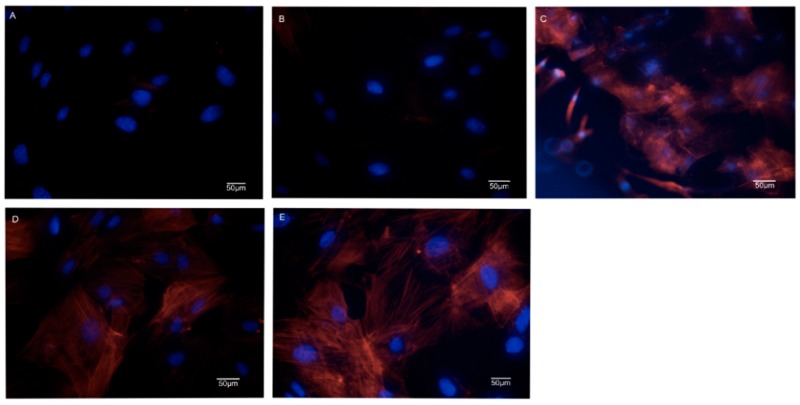
Confocal images of BMSCs on the SF/SPI-based scaffolds ((**A**): SF; (**B**): SF/SPI; (**C**): SF/SPI/GO; (**D**): SF/SPI/β-TCP; (**E**): SF/SPI/GO/β-TCP).

**Figure 12 polymers-12-00069-f012:**
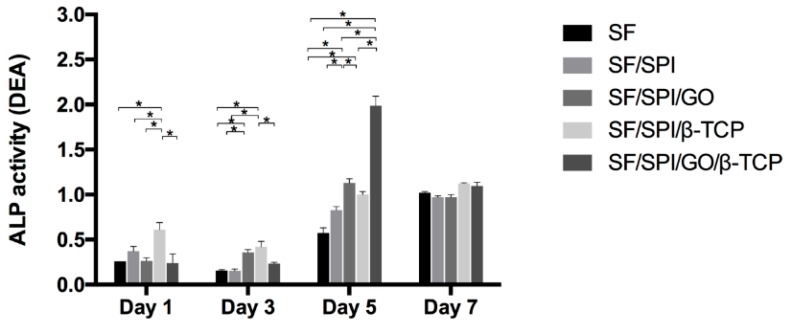
ALP activities of BMSCs on the SF/SPI-based composite scaffolds on day 1, 3, 5 and 7. Statistical significance relative to each group: * *p* < 0.05.

**Figure 13 polymers-12-00069-f013:**
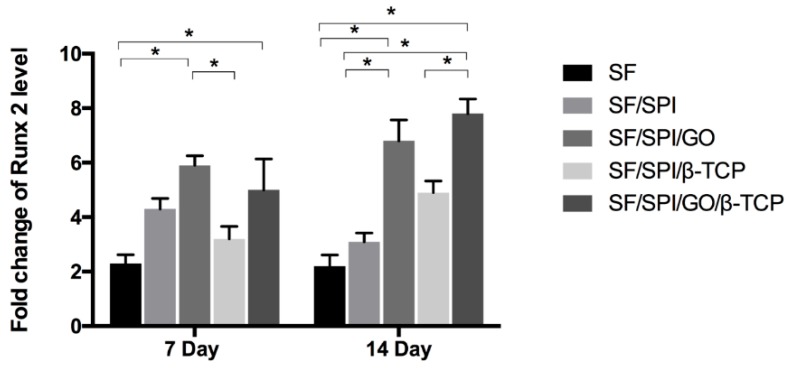
Runx 2 mRNA expression of BMSCs on the SF/SPI-based composite scaffolds on day 7 and 14. Statistical significance relative to each group: * *p* < 0.05.

**Figure 14 polymers-12-00069-f014:**
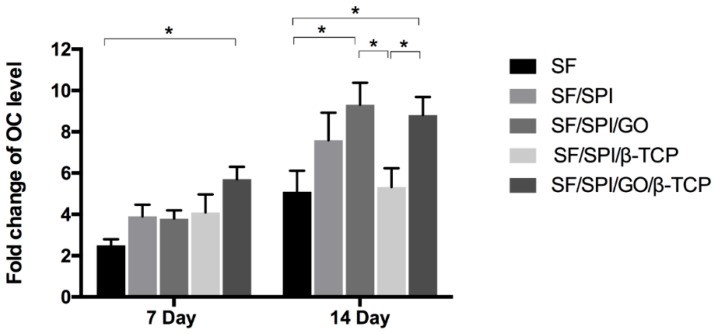
OC mRNA expression of BMSCs on the SF/SPI-based composite scaffolds on day 7 and 14. Statistical significance relative to each group: * *p* < 0.05.

**Figure 15 polymers-12-00069-f015:**
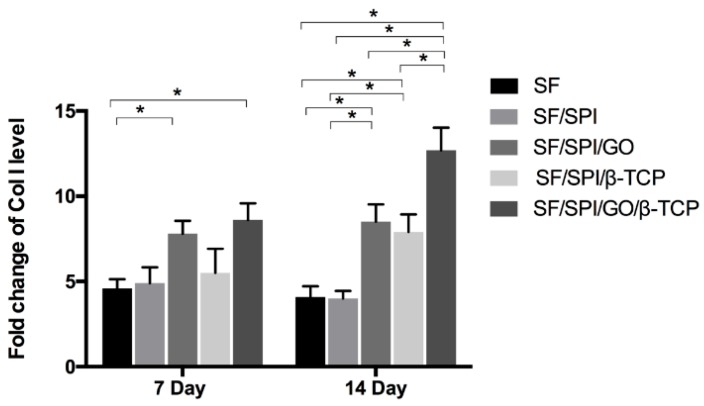
Col I mRNA expression of BMSCs on the SF/SPI-based composite scaffolds on day 7 and 14. Statistical significance relative to each group: **p* < 0.05.

**Table 1 polymers-12-00069-t001:** Primers for qRT-PCR.

Genes	Forward Primer	Reverse Primer
Runx2	CGCCTCACAAACAACCACAG	TCACTGTGCTGAAGAGGCTG
OC	CATGAAGGCTTTGTCAGACT	CTCTCTCTGCTCACTCTGCT
Col I	CCACCCCAGGGATAAAAACT	GGAGAGGAGTGCCAACTCCAG
GAPDH	AGTGCCAGCCTCGTCTCATA	GATGGTGATGGGTTTCCCGT

**Table 2 polymers-12-00069-t002:** The pore size and porosity of SF-based composite scaffolds.

	SF	SF/SPI	SF/SPI/GO	SF/SPI/β-TCP	SF/SPI/β-TCP/GO
Pore size (μm)	117.76 ± 6.33	113.37 ± 6.33	108 ± 6.33	232.53 ± 4.09	194 ± 6.18
Porosity (%)	79.32 ± 1.62	82.28 ± 2.15	87.66 ± 2.77	82.63 ± 1.04	80.45 ± 2.04

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
