# Peer review of "Synergistic Effects on Incorporation of β-Tricalcium Phosphate and Graphene Oxide Nanoparticles to Silk Fibroin/Soy Protein Isolate Scaffolds for Bone Tissue Engineering"

_polymers, 2020, doi:10.3390/polym12010069_

Round 1
Reviewer 1 Report
In this article, authors incorporated beta-TCP and GO nanoparticles into the SF/SPI scaffolds that could invariably improve the physicochemical and biological properties, particularly, enhance osteo inductive ability of the SF/SPI scaffolds. The structural, physicochemical and biological properties of these SF/SPI-based composite scaffolds were evaluated and they showed the synergetic effect of their beta-TCP and GO incorporated scaffold. This article was well organized and sentences were cleared. If only a few changes are made, they will be published in `Polymers’.
1. In authors’ experiments, GO appears to be nontoxic. However, there are fundamental questions about the toxicity of the material, such as whether GO breaks through cell membranes and causes problems.
2. Morphologies using SEM could confirm the initial cell adhesion but not proliferation. Howeverm quantitative evaluation of cell proliferation is required, and this test is possible with absorbance testing using cck-8.
3. There is no explanation of the proliferation-related image in Figure 11. Additional explanation is needed.
4. There is a problem of weakening the strength in freeze dried scaffold. In the mechanical test of Figure 3, only the group with beta-TCP was 1 MPa, and the strength of group with both beta-TCP and GO dropped to 0.8. The author said that the trabecular bone is 1 ~ 7MPa, so considering the result of group with beta-TCP and GO, isn't it weak?
5. What experimental condition is the ‘control’ in Figure 10? It needs explanation in the manuscript.
6. In Figure 10, the cell viability in the SF scaffold continues to drop for whole experiment time at 85% of day 1, and the authors confirmed. However, because silk fibrin is a very good biocompatible material, this result conflict with an existing knowledge. The experiment is likely to be Even if the experiment is successful, an explanation of the results is necessary.
7. In figures 13~15, Runx2, Col I results of 7 days and OC result of 14 days were similar. And Runx2, Col I results of 14 days were similar. Observation of 21 days or longer period of OC seems to show the synergy effect of the simultaneous use of beta-TCP and GO.
8. The three colors of light grey, dark grey, and black in Figure 4 do not match graphs with explanatory remarks. It is recommended to change the color to readability.
Author Response
Comments: In this article, authors incorporated beta-TCP and GO nanoparticles into the SF/SPI scaffolds that could invariably improve the physicochemical and biological properties, particularly, enhance osteo inductive ability of the SF/SPI scaffolds. The structural, physicochemical and biological properties of these SF/SPI-based composite scaffolds were evaluated and they showed the synergetic effect of their beta-TCP and GO incorporated scaffold. This article was well organized and sentences were cleared. If only a few changes are made, they will be published in `Polymers’.
Response: Thanks a lot for the faithful comments!
Comment 1: In authors’ experiments, GO appears to be nontoxic. However, there are fundamental questions about the toxicity of the material, such as whether GO breaks through cell membranes and causes problems.
Response: This is a very good question. In this study, the GO nanoparticles are incorporated in the SF/SPI-based composite scaffolds through solution mixture and they are encapsulated in the crosslinked protein chains tightly, the toxicity has not been exposed within the limited in vitro culture periods. Nevertheless, we will pay attention to this issue in the future studies with increased in vitro culture times as well as component analyses.
Comment 2: Morphologies using SEM could confirm the initial cell adhesion but not proliferation. Howeverm quantitative evaluation of cell proliferation is required, and this test is possible with absorbance testing using cck-8.
Response: Yes, quantitative evaluation of cell proliferation is an important index for biocompatibilities of biomaterials. We have used the absorbance testing of CCK-8 for cell viability testing and the results are shown in Figure 10.
Comment 3: There is no explanation of the proliferation-related image in Figure 11. Additional explanation is needed.
Response: The explanation of the proliferation-related images in Figure 11 has been added (page 12, line 329-338).
Comment 4: There is a problem of weakening the strength in freeze dried scaffold. In the mechanical test of Figure 3, only the group with beta-TCP was 1 MPa, and the strength of group with both beta-TCP and GO dropped to 0.8. The author said that the trabecular bone is 1 ~ 7MPa, so considering the result of group with beta-TCP and GO, isn't it weak?
Response: For large bone tissue repair, the mechanical strength is very important for stress-loading. The strength of group with both beta-TCP and GO is weak. For small bone tissue repair, the strength of 0.8 MPa is okay.
Comment 5: What experimental condition is the ‘control’ in Figure 10? It needs explanation in the manuscript.
Response: The controls in Figure 10 are added in the Materials and Methods section (page 4, line 161-162).
Comment 6: In Figure 10, the cell viability in the SF scaffold continues to drop for whole experiment time at 85% of day 1, and the authors confirmed. However, because silk fibrin is a very good biocompatible material, this result conflict with an existing knowledge. The experiment is likely to be Even if the experiment is successful, an explanation of the results is necessary.
Response: The explanation has been added (page 11, 321-322).
Comment 7: In figures 13~15, Runx2, Col I results of 7 days and OC result of 14 days were similar. And Runx2, Col I results of 14 days were similar. Observation of 21 days or longer period of OC seems to show the synergy effect of the simultaneous use of beta-TCP and GO.
Response: Yes, they are. Longer periods of in vitro cultures are needed to show the synergy effect of the simultaneous use of beta-TCP and GO. At present, these are all the data we have.
Comment 8: The three colors of light grey, dark grey, and black in Figure 4 do not match graphs with explanatory remarks. It is recommended to change the color to readability.
Response: The three colors of light grey, dark grey, and black in Figure 4 have been changed to be readable.

Reviewer 2 Report
I commend the authors for their research on incorporating bTCP and GO into SF/SPI scaffolds for enhanced bone tissue engineering properties. Introduction was brief, all material and methods used were thoroughly explained. Graphs, figures, tables and all the results were well presented. Conclusions drawn from the results were logical. I would suggest publishing the article at its current form.
Author Response
Comment: I commend the authors for their research on incorporating bTCP and GO into SF/SPI scaffolds for enhanced bone tissue engineering properties. Introduction was brief, all material and methods used were thoroughly explained. Graphs, figures, tables and all the results were well presented. Conclusions drawn from the results were logical. I would suggest publishing the article at its current form.
Response: The comments are sincere. It is greatly appreciated.
